# Influence Patterns for Explaining Information Flow in BERT

**Kaiji Lu,** *Zifan Wang, Piotr Mardziel, Anupam Datta**
Electrical and Computer Engineering
Carnegie Mellon University
Mountain View, CA 94089

## Abstract

While *"attention is all you need"* may be proving true, we do not know *why*: attention-based transformer models such as BERT are superior but how information flows from input tokens to output predictions are unclear. We introduce *influence patterns*, abstractions of sets of paths through a transformer model. Patterns quantify and localize the flow of information to paths passing through a sequence of model nodes. Experimentally, we find that significant portion of information flow in BERT goes through skip connections instead of attention heads. We further show that consistency of patterns across instances is an indicator of BERT's performance. Finally, we demonstrate that patterns account for far more model performance than previous attention-based and layer-based methods.

## 1  Introduction

Previous works show that transformer models such as BERT [7] encode various linguistic concepts [24, 40, 13], some of which can be associated with internal components of each layer, such as internal embeddings or attention weights [24, 40, 15, 33]. However, exactly how information flows through a transformer from input tokens to the output predictions remains an open question. Recent attempts to answer this question include using attention-based methods, where attention weights are used as indicators of flow of information [5, 2, 49], or layer-based approaches[15, 14, 33], which identify important network units in each layer.

In this paper, we examine the information flow question through an alternative lens of gradient-based attribution methods. We introduce *influence patterns*, abstractions of sets of gradient-based paths through a transformer's entire computational graph. We also introduce a greedy search procedure for efficiently and effectively finding patterns representative of concept-critical information flow. Figure 1 provides an example of an influence pattern in BERT. We conduct an extensive empirical study of influence patterns for several NLP tasks: Subject-Verb Agreement (SVA), Reflexive Anaphora (RA), and Sentiment Analysis (SA). Our findings are summarized below.

- A significant portion of information flows in BERT go through skip connections and not attention heads, indicating that attention weights[2] alone are not sufficient to characterize information flow. In our experiment, we show that on average, important information flow through skip connections 3 times more often than attentions.

- By visualizing the extracted patterns, we show how information flow of words interact inside the model and BERT may use grammatically incorrect cues to make predictions.

- The consistency of influence patterns across instances of a task reflects BERT's performance for that task.

---

*Correspondence to `kaijil@andrew.cmu.edu`

35th Conference on Neural Information Processing Systems (NeurIPS 2021).

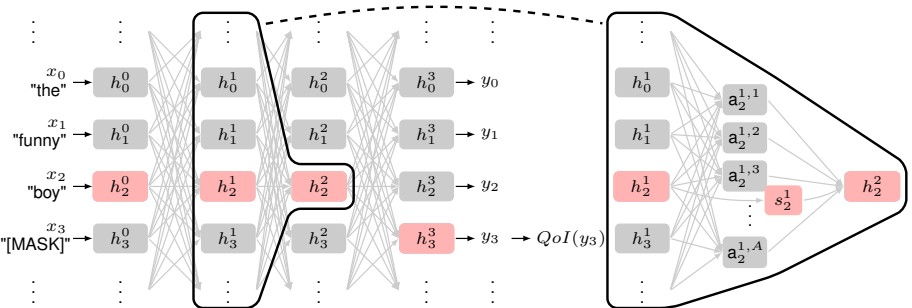

Figure 1: BERT architecture (left) and details of a transformer layer (right) for an instance of the SVA task, which evaluates whether the model chooses the correct verb form `is` over `are` for [MASK] to agree with the subject. An example of a pattern is highlighted with red nodes. $x$ and $h$ are input and internal embeddings, $a$ attention heads, $y$ output logits, $QoI$ the function for computing quantity of interest.

- Through ablation experiments, we find that influence patterns account for information flows in BERT on average 74% and 25% more accurately than prior attention-based and layer-based explanation methods[2, 23, 9], respectively.

## 2 Background

We begin this section by introducing notations and architecture of BERT in Sec. 2.1. We then introduce *distributional influence* as an axiomatic method to explain the output behavior of any deep model in Sec. 2.2, which serves as a building block for our method to follow in the next section.

### 2.1 BERT

Throughout the paper we use $\mathbf{x}$ to denote a vector and $|S|$ to denote the cardinality of a set $S$ or number of nodes in a graph $S$. We begin with the basics of the BERT architecture [7, 44] (presented in Fig. 1). Let $L$ be the number of layers in BERT, $H$ the hidden dimension of embeddings at each layer, and $A$ the number of attention heads. The list of input word embeddings is $\mathbf{x} \overset{\text{def}}{=} [\mathbf{x}_1, \mathbf{x}_2, ..., \mathbf{x}_N], \mathbf{x}_i \in \mathbb{R}^d$. We denote the output of the $l$-th layer as $\mathbf{h}^l_{1:N}$. First layer inputs are $\mathbf{h}^0_{1:N} \overset{\text{def}}{=} \mathbf{x}_{1:N}$. We use $a^{l,i}_j$ to denote the $j$-th attention head from the $i$-th embedding at $l$-th layer and $s^l_j$ to denote the skip connection that is "copied" from the input embedding from the previous layer then combined with the attention output. The output logits are denoted by $\mathbf{y}$.

**Computation Graph of BERT** A deep network can be viewed as a computational graph $\mathcal{G} \overset{\text{def}}{=} (\mathcal{V}, \mathcal{F}, \mathcal{E})$, a set of nodes, activation functions, and edges, respectively. In this paper, we assume the graph is directed, acyclic, and does not contain more than one edge per adjacent pair of nodes. A path $p$ in $\mathcal{G}$ is a sequence of graph-adjacent nodes $[p_1, p_2, \cdots, p_t]$; $p_t$ is the output node. Thus, the Jacobian passing through a path $p$ evaluated at input $\mathbf{x}$ is $\prod^{-1}_{i=1} \partial p_i(\mathbf{x})/\partial p_{i-1}(\mathbf{x})$ as per chain rule. We further denote the Jacobian of the output of node $n_i$ w.r.t the output of connected (not necessarily directly) predecessor node $n_j$ evaluated at $\mathbf{x}$ as $\partial n_j(\mathbf{x})/\partial n_i(\mathbf{x})$.

For computational and interpretability reasons, an ideal graph would contain as few nodes and edges as possible while exposing the structures of interest. For BERT we propose two graphs: *embedding-level graph* $\mathcal{G}_e$ corresponding to the nodes and edges shown in Fig. 1 (left) to explain how the influence of input embeddings flow from one layer of internal representations to another and to the eventual prediction; and *attention-level graph* $\mathcal{G}_a \supset \mathcal{G}_e$ that additionally includes attention head nodes and skip connection nodes as in Fig. 1 (right), a finer decomposition to demonstrate how influence from the input embedding flows through the attention block (or skip connections) within each layer.

## 2.2  Explaining Deep Neural Networks

Gradient-based explanations [39, 9, 37] are well-studied in explaining the behavior of a deep model by attributing the model's output to each input feature as its feature importance score. While existing approaches often explain with the most important features for a single class $i$, *distributional Influence* [23] generalizes gradient-based approaches to answer a broader set of questions, e.g. *why `[MASK]` should be `IS` instead of `ARE`* (Figure 1), by introducing *quantity of interest* (QoI). Suppose a general network $f : \mathbb{R}^d \to \mathbb{R}^n$, a QoI is a differentiable function $q(f(x))$ that outputs a scalar to incorporate the subject of an explanation. For example, to answer the aforementioned question, we can define $q(f(x)) = f(x)_{\texttt{IS}} - f(x)_{\texttt{ARE}}$ where $f(x)_*$ is the logit output of class $*$. Formally, we introduce *Distributional Influence*.

**Definition 1 (Distributional Influence)**  *Given a model $f : \mathbb{R}^d \to \mathbb{R}^n$, an input $\mathbf{x}$, a user-defined distribution $\mathcal{D}(\mathbf{x})$, and a user-defined QoI $q$, Distributional Influence $g(\mathbf{x}; q, \mathcal{D})$ is defined as:*

$$g(\mathbf{x}; q, \mathcal{D}) \overset{\text{def}}{=} \mathbb{E}_{\mathbf{z} \sim \mathcal{D}(\mathbf{x})} \frac{\partial q(f(\mathbf{z}))}{\partial \mathbf{z}}$$

**Remark 1**  *Distributional Influence also leverages a user-defined distribution $\mathcal{D}(\mathbf{x})$ to capture the network's behavior in a neighborhood of the input of interest $\mathbf{x}$. By introducing $\mathcal{D}(\mathbf{x})$, we can prove that several popular attribution methods are specific cases of the distributional influence; e.g. when $\mathcal{D}(\mathbf{x})$ is a Gaussian Distribution, $g(\mathbf{x}; q, \mathcal{D})$ reduces to Smooth Gradient[37]; when $\mathcal{D}(\mathbf{x})$ is a uniform distributions over a path $c = \{\mathbf{x} + \alpha(\mathbf{x} - \mathbf{x}_b), \alpha \in [0, 1]\}$ from a user-defined baseline input $\mathbf{x}_b$ to $\mathbf{x}$, $\mathcal{D}(\mathbf{x})$ reduces to Integrated Gradient [37].*

We use $\mathcal{D}(\mathbf{x})$ as the uniform distribution over a linear path described in Remark 1 in the rest of the paper because it provides several provable properties[39] to ensure the faithfulness of our explanations. We approximate the expectation in Def. 1 by sampling discrete points in the uniform distribution.

## 3  Tracing Influence Flow with Patterns

To explain how different concepts in the input flow to final predictions in BERT, it is important to show how the information from each input word flows through each intermediate layer and finally reaches the output embedding of interest, e.g. `[MASK]` for pretraining or `[CLS]` for fine-tuned models. Prior approaches have use the attention weights to build directed graphs from one embedding to another [2, 49]. These approaches use heuristics to treat high attention weights as indicators of important information flow between layers. However, as more work starts to highlight the axiomatic justifications of gradient-based methods over attentions weights as an explanation approach [42], we therefore explore an orthogonal direction in applying distributional influence in BERT to trace the information flow. Since distributional influence only attributes the output behavior over the input features, in this section, we generalize it to find important internal components that faithfully account for the output behavior.

**Tracing Influence by Patterns**    By viewing BERT as a computation graph ($\mathcal{G}_e$ or $\mathcal{G}_a$ in 2.1), we restate the problem: given a source node $s$ and a target node $t$, we seek a significant pattern of nodes from $s$ to $t$ that shows how the influence from $s$ traverses from node to node and finally reaches $t$. An exhaust way to rank all paths by the amount of influence flowing from $s$ to $t$ is possible in smaller networks, as is done by Lu et al. [27]. However, the similar approach lacks scalability to large models like BERT. Therefore, we propose a way to greedily narrow down the searching space from all possible paths to specific *patterns*. That is, our approach is two-fold: 1) we employ abstractions of sets of paths as the localization and influence quantification instrument; 2) we discover influential patterns with a greedy search procedure that refines abstract patterns into more concrete ones, keeping the influence high. We begin with the formal definition of a *pattern*.

**Definition 2 (Pattern)**  *A pattern $\pi$ is a sequence of nodes $[\pi_1, \pi_2, \cdots, \pi_{-1}]$ such that for any pair of nodes $\pi_i, \pi_{i+1}$ adjacent in the sequence (not necessarily adjacent in the graph), there exists a path from $\pi_i$ and $\pi_{i+1}$.*

A pattern $\pi$ abstracts a set of paths, written $\gamma(\pi)$ that follow the given sequence of nodes but are free to traverse the graph between those nodes in any way. Interpreting paths and patterns as sets, we define

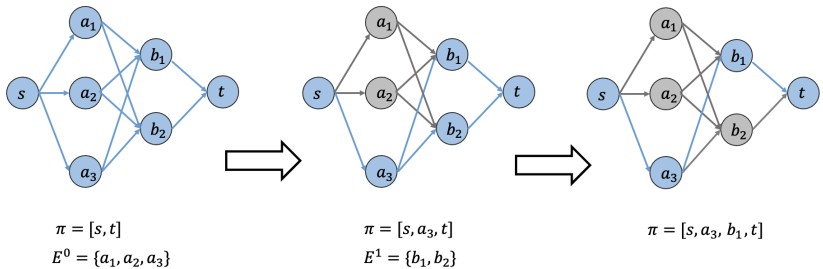

$$\pi = [s, t] \qquad\qquad \pi = [s, a_3, t] \qquad\qquad \pi = [s, a_3, b_1, t]$$
$$E^0 = \{a_1, a_2, a_3\} \qquad\qquad E^1 = \{b_1, b_2\}$$

Figure 2: A visual illustration of Guided Pattern Refinement (GPR) in a toy example. We start with a pattern $\pi = [s, t]$ containing only the source and the target node. At each step we define a guided set $E^0$ and $E^1$, respectively and find the node in the guided set that maximize the pattern influence (Def. 3). GPR finally returns a pattern $\pi = [s, a_3, b_1, t]$ that abstracts a single path.

$\gamma(\pi) \stackrel{\text{def}}{=} \{p \subseteq \mathcal{P} : \pi \subseteq p\}$ where $\mathcal{P}$ is the set of all paths from $\pi_1$ to $\pi_{-1}$. If every sequence-adjacent pair of nodes is directly connected then the pattern abstracts a single path. To quantify the amount of information that flows from the input node to the target node over a particular pattern, we propose *Pattern Influence*, motivated by distributional influence.

**Definition 3 (Pattern influence)** *Given a computation graph and a user-defined distribution $\mathcal{D}$, the influence of an influence pattern $\pi$, written $\mathcal{I}(\mathbf{x}, \pi)$ is the total influence of all the paths abstracted by the pattern:* $\mathcal{I}(\mathbf{x}, \pi) \stackrel{\text{def}}{=} \sum_{p \in \gamma(\pi)} \mathbb{E}_{\mathbf{z} \sim \mathcal{D}(\mathbf{x})} \prod_{i=1}^{-1} \frac{\partial p_i(\mathbf{z})}{\partial p_{i-1}(\mathbf{z})}.$

**Proposition 1 (Chain Rule)** $\mathcal{I}(\mathbf{x}, \pi) = \mathbb{E}_{\mathbf{z} \sim \mathcal{D}(\mathbf{x})} \prod_{i=1}^{-1} \frac{\partial \pi_i(\mathbf{z})}{\partial \pi_{i-1}(\mathbf{z})}$ *for any distribution $\mathcal{D}(\mathbf{x})$.*

Prop. 1 (proof in Appendix A) simplifies the evaluation of the pattern influence by specifying the exact set of internal nodes through which influence flows in a computational graph. We hereby introduce a greedy way of finding the most influential pattern in both $\mathcal{G}_e$ and $\mathcal{G}_a$.

**Guided Pattern Refinement(GPR)** Starting with source and target nodes $s$ and $t$ along with a initialized pattern $\pi^0 = \{s, t\}$ representing all paths between $s$ and $t$, we construct $\pi^1$ by adding one[2] sequence-adjacent node $e^0$ (there is a direct path between $s$ and $e^0$) from a *guiding set* $E^0$ that maximizes the influence of the resulting pattern such that:

$$e^0 = \arg\max_{e \in E^0} \{\mathcal{I}(\mathbf{x}, [s, e, t]), \pi^1 = [s, e^0, t], \tag{1}$$

This procedure is iterated until we exhaust the last guiding set. We show an example of GPR in a toy graph in Fig. 2. For an embedding-level graph $\mathcal{G}_e$, each guiding set $E^l$ includes all embeddings $\mathbf{h}_{1:N}^l$ at layer $l$. For the attention-level graph $\mathcal{G}_a$, we refine on the embedding-level pattern $\pi^e$ by only expanding $\pi^e$ from addition nodes in $\mathcal{G}_a$ compared with $\mathcal{G}_e$: we perform GPR iterations between $\pi_l^e, \pi_{l+1}^e \in \pi^e$ with the guiding set $E_a^l$ which includes $A$ attention heads $\mathbf{a}^l$ and the skip node $\mathbf{s}^l$(Fig. 1 Right), until we reach the same last node in $\pi^e$. The returned attention-level pattern $\pi^a$ thus abstracts a single path from the source to the target in $\mathcal{G}_a$. As the attention-level analysis refines the embedding-level analysis, the produced attention-level pattern $\pi^a$ abstracts a strict subset of the paths of the attention-level graph that the embedding-level pattern $\pi^e$ abstracts. That is, $\pi^e \subset \pi^a$ while $\gamma(\pi^a) \subset \gamma(\pi^e)$. The detailed algorithm of GPR and analysis of its optimality can be found in Appendix B.1 and B.2.

## 4  Experiment

Experiments in this section demonstrate our method as a tool for interpreting end-to-end information flow in BERT. Specific visualizations exemplify these interpretations including the importance of skip connections and BERT's encoding of grammatical concepts are included in Sec. 4.2. Sec. 4.3

---

[2]The algorithm can be easily adapted to include more nodes per layer. However, we found one node per layer retain a reasonable proportion of influence for the tasks evaluated in this paper(See Sec.4.4).

explores the consistency of patterns across instances and template positions and how they relate to task performances and influence magnitudes in. Sec 4.4 demonstrates two advantages of patterns in explaining the information flow of BERT over baselines: 1) abstracted patterns, with much fewer nodes compared to the whole model, carry sufficient information for the prediction. That is, without the information outside the refined pattern, the model shows no significant performance drop; 2) At the same time, patterns are sparse but concentrated in BERT's components.

## 4.1 Setup

**Tasks.** We consider two groups of NLP tasks: (1) *subject-word agreement (SVA)* and *reflexive anaphora (RA)*. We explore different forms of sentence stimuli(subtask) within each task: object relative clause (Obj.), subject relative clause (Subj.), within sentence complement (WSC), and across prepositional phrase (APP) in SVA [28]; number agreement (NA) and gender agreement (GA) in RA [24]. Both datasets are evaluated using masked language model (MLM) as is used in Goldberg [13]. We sample 1000 sentences from each subtask evenly distributed across different sentence types (e.g. singular/plural subject & singular/plural intervening noun) with a fixed sentence structure; (2) *sentiment analysis(SA)*: we use 220 short examples (sentence length$\leq 17$) from the evaluation set of the 2-class GLUE SST-2 sentiment analysis dataset [47]. More details and examples of each task can be found in Appendix D.1.

**Models.** For linguistic tasks, We evaluate our methods with a pretrained BERT($L = 6, A = 8$) [43], referred hereby as BERT$_{\text{SMALL}}$. For SST-2 we fine-tuned on the pretrained BERT$_{\text{BASE}}$[7] with $L = 12, A = 12$. The models are similar in sizes compared to the transformer models used in [2]. All computations are done with a Titan V on a machine with 64 GB of RAM. See Appendix D.2 for more details.

**Implementation of GPR.** Let the target node for SVA and RA tasks be the output of the QoI score $q(\mathbf{y}) \overset{\text{def}}{=} y_{correct} - y_{wrong}$. For instance, $y_{\text{IS}} - y_{\text{ARE}}$ for the sentence she [MASK] happy. Similarly, we use $y_{\text{positive}} - y_{\text{negative}}$ for sentiment analysis. We choose an uniform distribution over a linear path from $\mathbf{x}_b$ to $\mathbf{x}$ as the distribution $\mathcal{D}$ in Def. 3 where the $\mathbf{x}_b$ is chosen as the the input embedding of [MASK] because it can viewed a word with no information. For a given input token $\mathbf{x}_i$, we apply GPR differently depending on the sign of distributional influence $g(\mathbf{x}; q, \mathcal{D})$: if $g(\mathbf{x}; q, \mathcal{D}) \geq 0$, we maximize the pattern influence towards $q(\mathbf{y})$ at each iteration of the GPR otherwise we maximize pattern influence towards $-q(\mathbf{y})$. We use $\pi_i$ as the extracted patterns for individual input word $i$. When explaining the whole input sentence, we collect all refined patterns for each word and use $\Pi = \bigcup_i \pi_i$. Further, we denote $\Pi_+$ as the set of patterns for all positively influential words. Both terms may be further decorated by $a$ or $e$ to denote attention-level or embedding-level results.

## 4.2 Visualizing Influence Patterns

This section does not serve as an evaluation but an exploration of insightful results and succinct conclusions an human user can learn from our proposed technique. We visualize the information flow identified by patterns found by GPR, and compare with those generated by attention weights as explored in the literature [2, 49][3]. We therefore use $\Pi_{\text{attn}}$ to denote a pattern of nodes by maximizing the product of the corresponding attention weights between each pairs of nodes from adjacent layers. Implementation details are included in Appendix C.1.

We first focus on instances of the subtask *SVA across object relative clauses (SVA-obj)*, which are generated from the template: the SUBJECT that the ATTRACTOR VERB [MASK](is/are) ADJ.. We observe in Figure 3 that the subject words exert positive input influences on the correct choice of the verb, and the intervening noun (attractor) exerts negative influence, which is true for both $\mathcal{I}(\mathbf{x}_i, \pi_i^e)$ and $\mathcal{I}(\mathbf{x}_i, \pi_i^a)$ (blue and purple bars in Figure 3a and 3b). While $\Pi_{\text{attn}}$(Fig. 3c) does not distinguish between positively influential words and negative ones, nor do they show an interpretable pattern. Our other main findings are discussed as follows.

**Finding I: Skip Connection Matters.** Horizontal dashed lines in Figure 3 indicate that influence can flow through layers at the same word position via skip connections, which is not isolated as separate nodes $\Pi_{\text{attn}}$ (shown in Fig. 3c). Fig. 3a and 3b also show that the influence from subject

---

[3]The two works compute attribution from input to internal embeddings by aggregating attention weights(averaged across heads) across layers while attribution to individual paths is implicit.

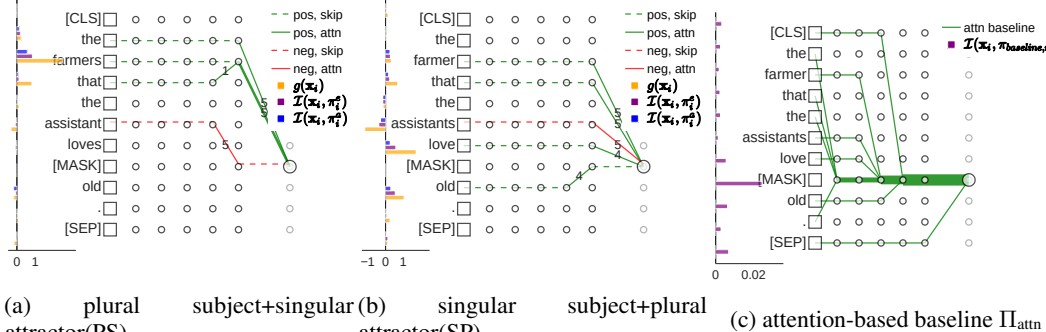

(a) plural subject+singular attractor(PS)

(b) singular subject+plural attractor(SP)

(c) attention-based baseline $\Pi_{attn}$

Figure 3: (a)(b) Patterns for two instances of *SVA-Obj*.(c)Baseline pattern $\Pi_{attn}$. For each plot: Left: bar plots of the distributional influence $g(\mathbf{x}_i)$(yellow), $\mathcal{I}(\mathbf{x}_i, \pi_i^e)$ (purple) and $\mathcal{I}(\mathbf{x}_i, \pi_i^a)$ (blue) (or $\mathcal{I}(\mathbf{x}_i, \pi_{baseline,i})$) for each word at position $i$. Right: Extracted patterns $\Pi$ from selective words. Square nodes and circle nodes denote input and internal embeddings, respectively. In (a) and (b), influence flowing through skip connections is represented by dashed lines and attention heads in solid lines; the edges are are marked with the corresponding attention head number (ranging from 1 to $A$) in $\Pi^a$. Line colors represent the sign of influence (red as negative and green as positive).

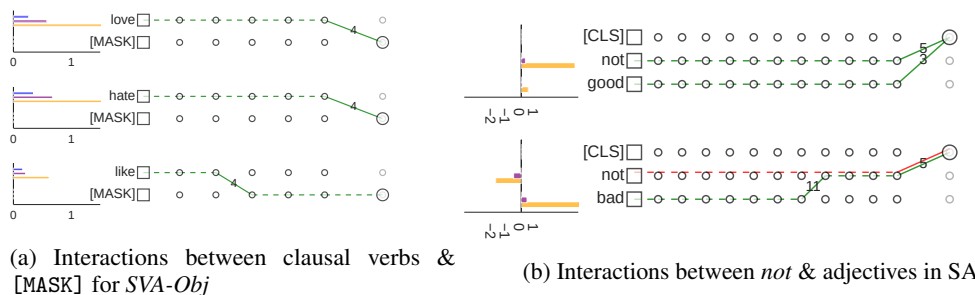

(a) Interactions between clausal verbs & `[MASK]` for *SVA-Obj*

(b) Interactions between *not* & adjectives in SA

Figure 4: (a) Patterns on three clausal verbs from SP, *SVA-obj*.(b) Patterns for two instances in the SA task. Legend follows Figure 3.

words `farmer[s]` travels through skip connections across layers before it transfers into the attention head 5 in the last layer, indicating the "number information" from the subject embedding flows directly to the output. Interestingly, this also explains why attentions can be pruned effectively without compromising the performance [21, 31, 46] as important information may not flow through attentions at all. Namely, they are simply "copied" to the next layer through the skip connection. In fact, attention heads are traversed far less often than skip connections, which account for $75.4\%$ of nodes in $\Pi_a$ across all tasks evaluated.

**Finding II:** `that` **as an Attractor.** Fig. 3a and 3b shows that the singular subject is less influential than the plural subject, especially when compared to the large negative influence from the attractor. Besides, the word `that` behaves like a singular pronoun in the singular subject case (Figure 3b), flowing through the same pattern as the subject (skip connections + attention head 5), whereas `that` is more like a grammatical marker (relativizer) in the plural subject case (Figure 3a): the pattern from `that` converge to the subject in the second to last layer via a different attention head 1. An explanation to the observed difference is that `that` can either be used as a singular pronoun in English, or a marker that encodes the syntactic boundary of the clause to help identify the subject and ignore the attractor. This finding reveals that although the latter one is always the correct usage for `that` across all instances, BERT may resort to the "easier"(while ungrammatical) encoding of `that` as a pronoun when the number happens to be singular.

**Finding III:** `love`, `hate` **and** `like` Figure 4a shows the pattern across three instances containing different clausal verbs by replacing `love` in Figure 3b with `like` or `hate`. We observe that `hate` and `love` are more influential, with a distinct and more concentrated pattern compared to that of

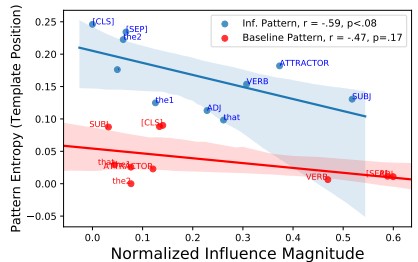

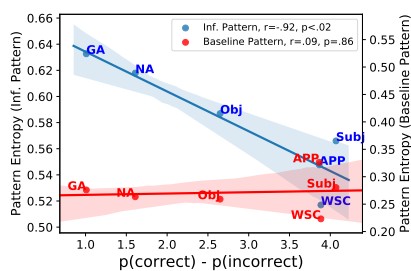

(a) Pattern Entropy vs. Average Influence Magnitude for template positions in *SVA-Obj*

(b) Pattern Entropy vs. Average QoI for linguistic tasks (SVA & RA)

Figure 5: Relationship between Task Performance, Influence Magnitude and Pattern Entropy

the word `like`, while one comparable to that of the subject or noun attractor. Therefore, `hate` and `love` are treated more like singular nouns than `like`, contributing positively to the correct prediction of the verb's number (but for the wrong grammatical reason). This discrepancy also corroborates different accuracy for instances in the three subsets: `hate`: .99; `love`: 1.0; `like`: .82. In fact, the number of misclassifications containing `like` accounts for more than 33% of all misclassifications in *SVA-Obj(SP)*.

**Finding IV: Interactions with `not`.** Two instances of sentiment analysis which BERT classifies correctly (negative for `not good.` and positive for `not bad.`) in Fig. 4b shows that: 1) in `not good.`, `not` and `good` does not interact internally and `not` carries much more positive influence than `good` towards the correct class (negative sentiment); 2) in `not bad.`, on the contrary, `not` contributes negatively to the correctly class (positive sentiment) and `bad` interacts with the `not` in an internal layer and exhibit a large positive influence. This disparity shows the model may treat `not` as a word with negative sentiment by default, and only when the subsequent adjective is negative, can it be used as a negation marker to encode the correct sentiment.

The asymmetries and oddities of patterns of Finding II to IV show how BERT may use ungrammatical correlations for its predictions. We include more examples of patterns in Appendix E.

## 4.3 Consistency of Patterns Across Instances

The prior section offered examples building connections between information flow and patterns. Since patterns are computed for each word of each instance individually, a global analysis of consistency/variation of patterns across instances can help us make deeper insights and make more general conjectures. We quantify this variation with *pattern entropy* on the linguistic tasks (SVA & RA) with fixed templates, where for each instance, each template position are chosen from a group of words. Interpreting the collection of instances, or template positions, in a task as a distribution, pattern entropy is the entropy of the binary probability that a node is part of a pattern, averaged over all nodes (minimally this is 0 if all patterns incorporate the exact same set of nodes). With a node $n$ in graph $G$, a set of $K$ instances, $\pi_k$ as the pattern of instance $k$ by abuse of notation and $H$ as the entropy function: pattern entropy of a pattern $\pi$ is defined as:

$$\hat{H}(\pi, G) = \frac{\sum_{n \in G} H(p_n(\pi))}{|G|}; p_n(\pi) = \frac{\sum_k \mathbb{1}(n \in \pi_k)}{K}$$

Figure 5a shows an inverse linear relation between influence magnitude and the pattern entropy of each template position($\pi_i$) in *SVA-Obj*. For example, the subject position has on average high influence magnitude and low pattern entropy, while grammatically unimportant template positions such as `[CLS]` and the period token has relatively low influence and high entropy. Note that this fit is not perfect: attractors, for example, have high influence magnitude but also high entropy's due to their disparate behaviors among instances shown in Sec 4.2.

Figure 5b demonstrates a more global inverse relation between the pattern entropy of $\Pi_+$ and task-specific performance(average QoI) for 6 linguistic subtasks studied. This indicates that the more consistent a pattern is for a concept, i.e, the more consistent how a model locates the positively

| Patterns | SVA | | | | RA | | SA |
|---|---|---|---|---|---|---|---|
| | Obj. | Subj. | WSC | APP | NA | GA | |
| $\Pi_+^e$ (ours) | **.96** | **.99** | **1.0** | **.96** | **.98** | **.99** | **.97** |
| $\Pi_{rand}^e$ | .55 | .60 | .55 | .55 | .55 | .56 | .52 |
| $\Pi_{attn}^e$ | .61 | .55 | .49 | .63 | .56 | .65 | .47 |
| $\Pi_{cond}^e$ | .93 | .91 | .88 | .90 | .71 | .95 | .94 |
| $\Pi_{inf}^e$ | .66 | .71 | .50 | .56 | .68 | .50 | .49 |
| $\Pi_+^a$ (ours) | **.70** | **.62** | **.56** | **.65** | **.78** | **.89** | **.86** |
| $\Pi_{rand}^a$ | .50 | .50 | .50 | .50 | .50 | .50 | .49 |
| $\Pi_{repl\_skip}^a$ | .50 | .58 | .54 | .52 | .55 | .50 | .48 |
| original | .96 | 1.0 | 1.0 | 1.0 | .83 | .73 | .92 |

| Metrics | SVA | | | | RA | | SA |
|---|---|---|---|---|---|---|---|
| | Obj. | Subj. | WSC | APP | NA | GA | |
| conc.($\Pi_+^e$) | .33 | .30 | .27 | .33 | .31 | .22 | .07 |
| conc.($\Pi_-^e$) | .29 | .30 | .38 | .31 | .31 | .23 | .05 |
| share($\Pi_+^e$) | .06 | .09 | .03 | .06 | .04 | .02 | .01 |
| conc.($\Pi_+^a$) | .21 | .18 | .16 | .18 | .18 | .14 | .01 |
| conc.($\Pi_-^a$) | .17 | .16 | .25 | .15 | .17 | .15 | .01 |
| share($\Pi_+^a$) $(\cdot 10^{-6})$ | 1.8 | 1.7 | 1.6 | 1.5 | 1.3 | 1.3 | 3e-2 |
| Alignment Rate | .58 | .57 | .57 | .59 | .46 | .53 | .34 |

Table 1: Summary of quantitative results, with $* \in \{e, a\}$, denoting embedding & attention-level metrics. Left: Ablated accuracies of $\Pi_+$ and baseline patterns. Right: Sparsity metrics; conc.($\Pi_\pm^*$): positive/negative *concentration*; share($\Pi^*$): *path share* of pattern $\Pi^*$.

influential signals, the better the model is at capturing that concept. In both figures, the baseline attention-based patterns ($\Pi_{attn}$) does not indicate this relation. Together with Finding II to IV in Sec. 4.2, it demonstrates that patterns may offer alternative insights in formalizing and answering high-level questions such as to what extent does BERT generalize to correct and consistent grammatical rules or use spurious correlations which varies across instances[30, 29, 34].

## 4.4 Evaluating Influence Patterns

This section serves as a formal evaluation of our proposed methods against existing baselines in tracing information flows. Except the attention baseline $\Pi_{attn}$ defined in Sec. 4.2, we also include the following baselines (implementation details to follow in Appendix C.1):

■ $\Pi_{rand}$: patterns with randomly sampled nodes.

■ $\Pi_{inf}$: patterns consisting of nodes that maximize the *internal influence* [23] for each guiding set.

■ $\Pi_{cond}$: patterns consisting of nodes that maximize the *conductance* [9] for each guiding set.

We compare various aspects of extracted patterns against the following baselines, including: (1) ablation experiments showing how influence patterns account for model performance; (2) the sparsity of the patterns in the computational graph. Finally, we also discuss the relation between patterns and attention-weights.

**Ablation.** We extend the commonly-used ablation study for evaluating explanation methods [8, 6, 3, 23, 27] from input features to internal patterns as a sanity check of our method. That is, we ablate BERT down to a simpler model: we only retain the nodes from $\Pi_+^e$ (or $\Pi_+^a$) while replacing other nodes by zeros. The retained and replaced nodes together are forward passed to the next layer with the original model parameters until a new set of nodes are retained or replaced. *Ablated Accuracy* denotes the accuracy of the ablated model. We show the results for our methods $\Pi_+^*$ and other baseline patterns[4] in Table 1. Each baseline retains the same number of nodes as the corresponding $\Pi_+$. Although this process seems to be highly invasive, the model ablated with $\Pi_+^*$ achieved the highest ablated accuracy uniformly over all tasks. All baseline patterns except $\Pi_{cond}^e$ [5] gives random guesses. For the attention-level graph $\Pi_a$, we also add a counterfactual pattern $\Pi_{repl\_skip}^a$ where we replace each skip node in $\Pi_+^a$ with its all $A$ corresponding attention heads. However, with much fewer nodes ablated, $\Pi_{repl\_skip}^a$ still produces random ablated accuracies. This corroborates Finding I in Sec. 4.2 that the skip connections relaying important information directly which attention block cannot replace. In summary, patterns refined by GPR account for the model's information flow more sufficiently compared to all baselines.

**Sparsity of Patterns.** A good explanation should not only account for the model performance, but also be sparse relative to the entire model semantics so as to be interpretable to humans. In this section, we quantify the sparsity of extracted patterns using two metrics: *path share* and *concentration*. With

---
[4] we run $\Pi_{rand}$ 50 times per task and average the ablated accuracy with no significant variation, the detailed statistics can be found in Appendix C.3

[5] We include discussions why $\Pi_{cond}^e$ has higher ablated accuracy than other baselines in Appendix C.2.

on average more than 10 words per sentence, the embedding and attention-level graphs contain at least $10^6$ and $10^{11}$ individual paths, respectively. The totality of these paths represent the entire semantics of the BERT model. *Path share*(i.e. $\text{share}(\Pi_+^*) \overset{\text{def}}{=} |\gamma_*(\Pi_+^*)|/|\mathcal{P}^*|$), is defined as the number of paths in a pattern over the total number of paths in the entire computational graph. *concentration*, on the other hand, is defined as the proportion of negative/positive pattern influence over the total positive/negative influence. It represents the average ratio between the blue/purple bars and the orange bars in Figure 3.

Table 1(Right) shows that the abstracted patterns contain only a small share of paths while accounting for a large portion of both positive and negative influence. In linguistic tasks, the embedding level abstracted pattern has a *concentration* around 0.3 ($\text{conc.}(\Pi_+^e)$), indicating that the input concept flows through single internal embeddings in each layer, instead of distributed to many words. Zooming in on the attention-level graph, a relative high concentration $\text{conc.}(\Pi_+^a)$, suggests that between the internal embeddings of adjacent layers, influence is also more concentrated to either one attention head or the skip connection. We speculate the much lower $\text{conc.}(\Pi_+)$ in SA is due to (1) the much larger model for the SA task with more layers and attention heads (2) sentiment information may be more diverse and complex than the information needed to encode syntax agreements, therefore input information may flow in a more distributed way. However, despite low concentration of pattern influence, the extracted patterns for SA are still effective in capturing the model performance as shown in Table 1(Left).

**Attention heads in $\Pi^a$ vs. Attention weight value.** Another observation is that the attention head nodes in $\Pi^a$ found by GPR aligns, to greater extent than random ($1/|A|$), with the head of the largest attention weight along the corresponding embedding-level edge, as shown by *alignment rate* in Table 1(Left). This partial alignment is expected since the Jacobian between two nodes correlates with the coefficients (weights) in the linear attention mechanism(product rule), while the opposite is also expected since (1) attention weights themselves are not fixed model parameters thus part of the gradient flow, (2) model components other than attention blocks come between two adjacent layers (e,g. dense layer & skip connections). In other words, attention weights are correlated, but not equivalent to gradient-based methods as explanations.

# 5   Related Work

Previous work has shown the encoding of syntactic dependencies such as SVA in RNN Language models [25, 16, 22, 19]. More extensive work has since been done on transformer-based architectures such as BERT. Probing classifiers has shown BERT encodes many types of linguistic knowledge[11, 14, 40, 41, 17, 20, 26, 24, 33, 15]. [13] discovers that SVA and RA in complex clausal structures is better represented in BERT compared to an RNN model.

A line of related works analyze self-attention weights of BERT[5, 45, 24, 12], where attention heads are found to have direct correspondences with specific dependency relations. [2] and [49], with which we compare in this paper, propose using attentions to quantify information flows in BERT. Attention weights as interpretation devices, however, have been controversial [36] and empirical analysis has shown that attention can be perturbed or pruned while retaining performance [21, 31, 46]. Our work demonstrates that attention mechanisms are only part of the computation graph, with each attention block complemented by other model components such as dense layer and skip connections; axiomatically justified influence patterns, however, can attribute to the whole computation graph. The strong influence passing through skip connections also corroborates the findings of [4] which finds input tokens mostly retain their identity. Besides pruning attentions, other works[32, 35, 18] also show that BERT is overparametrized and can be greatly compressed. Our work to some extent corroborates that point by pointing to the sparse gradient flow, while employing ablation studies only to verify the sufficiency of the extracted patterns.

A closely related and concurrent work[10] also shows the importance of skip connections and dense(MLP) layers by decomposing and analyzing forward-pass computations in self-attention modules. Our work, in comparison, introduces a gradient-based method that can be generalized to any model as long as they are differentiable. We also focus on how influential patterns can help us understand information flow of specific NLP tasks.

Recent work introducing influence paths [27] offers another form of explanation. Lu et al. [27] decomposed the attribution to path-specific quantities localizing the implementation of the given

concept to paths through a model. The authors demonstrated that for LSTM models, a single path is responsible for most of the input-output effect defining SVA. Directly applying individual paths to transformer-based models like BERT, however, results in an intractable number of paths to enumerate due to the huge number of computation edges in BERT.

# 6 Limitations and Future Work

We believe there are three limitations to address in future work. (1) GPR algorithm does not guarantee absolute optimality, however, we provide empirical evidence in support of the searching algorithm in Appendix B.2. (2) For interpretability reasons, we compute GPR by picking one node per guiding set, while more complicated information may distribute to multiple internal embeddings or attention heads. (3) The findings in Sec. 4.2 would benefit from more quantitative analysis to support more general claims(instead of speculations) on BERT's handling of various linguistic/semantic concepts. However, we will explore these limitations in future work and release our code and hope the proposed methods will serve as an insightful tool in future exploration.

# 7 Conclusion

We demonstrated influence patterns for explaining information flow in BERT. We highlighted the importance of skip connections and BERT's potentially mishandling of various concepts through visualized patterns. We inspected the relation between consistency of patterns across instances with model performance and quantitatively validated pattern's sufficiency in capturing model performance.

**Acknowledgement**    This work was developed with the support of NSF grant CNS-1704845. The U.S. Government is authorized to reproduce and distribute reprints for Governmental purposes not withstanding any copyright notation thereon. The views, opinions, and/or findings expressed are those of the author(s) and should not be interpreted as representing the National Science Foundation or the U.S. Government. We gratefully acknowledge the support of NVIDIA Corporation with the donation of the Titan V GPU used for this work.

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
