# A  Appendix: Proof of Proposition 1

**Proposition 1 (Chain Rule)** $\mathcal{I}(\mathbf{x}, \pi) = \mathbb{E}_{\mathbf{z} \sim \mathcal{D}(\mathbf{x})} \prod_{i=1}^{-1} \frac{\partial \pi_i(\mathbf{z})}{\partial \pi_{i-1}(\mathbf{z})}$ *for any distribution* $\mathcal{D}(\mathbf{x})$.

We first show that for a pattern $\pi = [\pi_1, \pi_1, ..., \pi_k, ..., \pi_n]$ the following equation holds:

$$\frac{\partial \pi_n}{\partial \pi_1} = \frac{\partial \pi_n}{\partial \pi_k} \frac{\partial \pi_k}{\partial \pi_1} \tag{2}$$

*Proof:* Let $\gamma(\pi_1 \to \pi_n)$ be a set of paths that the pattern $\pi$ abstracts. Therefore, we have

$$\frac{\partial \pi_n}{\partial \pi_1} = \sum_{p \in \gamma(\pi_1 \to \pi_n)} \prod_{i=1}^{-1} \frac{\partial p_i}{\partial p_{i-1}} \tag{3}$$

Similarly, we have

$$\frac{\partial \pi_k}{\partial \pi_1} = \sum_{p \in \gamma(\pi_1 \to \pi_k)} \prod_{i=1}^{-1} \frac{\partial p_i}{\partial p_{i-1}}, \quad \frac{\partial \pi_n}{\partial \pi_k} = \sum_{p \in \gamma(\pi_k \to \pi_n)} \prod_{i=1}^{-1} \frac{\partial p_i}{\partial p_{i-1}} \tag{4}$$

Suppose $|\gamma(\pi_1 \to \pi_k)| = N_1$, $|\gamma(\pi_k \to \pi_n)| = N_2$ and we denote $\prod_{i=1}^{-1} \frac{\partial p_i^{(j)}}{\partial p_{i-1}}$ as path gradient flowing from the path $j$. Therefore,

$$\frac{\partial \pi_n}{\partial \pi_1} = \sum_{p \in \gamma(\pi_1 \to \pi_n)} \prod_{i=1}^{-1} \frac{\partial p_i}{\partial p_{i-1}} \tag{5}$$

$$= \prod_{i=k}^{-1} \frac{\partial p_i^{(1)}}{\partial p_{i-1}} \cdot \prod_{i=1}^{k} \frac{\partial p_i^{(1)}}{\partial p_{i-1}} + \prod_{i=k}^{-1} \frac{\partial p_i^{(1)}}{\partial p_{i-1}} \cdot \prod_{i=1}^{k} \frac{\partial p_i^{(2)}}{\partial p_{i-1}} + ... + \prod_{i=k}^{-1} \frac{\partial p_i^{(1)}}{\partial p_{i-1}} \cdot \prod_{i=1}^{k} \frac{\partial p_i^{(N_1)}}{\partial p_{i-1}} \tag{6}$$

$$+ \prod_{i=k}^{-1} \frac{\partial p_i^{(2)}}{\partial p_{i-1}} \cdot \prod_{i=1}^{k} \frac{\partial p_i^{(1)}}{\partial p_{i-1}} + \prod_{i=k}^{-1} \frac{\partial p_i^{(2)}}{\partial p_{i-1}} \cdot \prod_{i=1}^{k} \frac{\partial p_i^{(2)}}{\partial p_{i-1}} + ... + \prod_{i=k}^{-1} \frac{\partial p_i^{(2)}}{\partial p_{i-1}} \cdot \prod_{i=1}^{k} \frac{\partial p_i^{(N_1)}}{\partial p_{i-1}} \tag{7}$$

$$+ ... \tag{8}$$

$$+ \prod_{i=k}^{-1} \frac{\partial p_i^{(N_2)}}{\partial p_{i-1}} \cdot \prod_{i=1}^{k} \frac{\partial p_i^{(1)}}{\partial p_{i-1}} + \prod_{i=k}^{-1} \frac{\partial p_i^{(N_2)}}{\partial p_{i-1}} \cdot \prod_{i=1}^{k} \frac{\partial p_i^{(2)}}{\partial p_{i-1}} + ... + \prod_{i=k}^{-1} \frac{\partial p_i^{(N_2)}}{\partial p_{i-1}} \cdot \prod_{i=1}^{k} \frac{\partial p_i^{(N_1)}}{\partial p_{i-1}} \tag{9}$$

$$= \sum_{j}^{N_2} (\prod_{i=k}^{-1} \frac{\partial p_i^{(j)}}{\partial p_{i-1}} \cdot \sum_{m}^{N_1} \prod_{i=1}^{k} \frac{\partial p_i^{(m)}}{\partial p_{i-1}}) \tag{10}$$

$$= \sum_{j}^{N_2} (\prod_{i=k}^{-1} \frac{\partial p_i^{(j)}}{\partial p_{i-1}}) \cdot \sum_{m}^{N_1} (\prod_{i=1}^{k} \frac{\partial p_i^{(m)}}{\partial p_{i-1}}) \tag{11}$$

$$= \sum_{p \in \gamma(\pi_1 \to \pi_k)} \prod_{i=1}^{-1} \frac{\partial p_i}{\partial p_{i-1}} \cdot \sum_{p \in \gamma(\pi_k \to \pi_n)} \prod_{i=1}^{-1} \frac{\partial p_i}{\partial p_{i-1}} \tag{12}$$

$$= \frac{\partial \pi_n}{\partial \pi_k} \frac{\partial \pi_k}{\partial \pi_1} \tag{13}$$

Now we prove Proposition 1.

$$\mathcal{I}(\mathbf{x}, \pi) = \sum_{p \in \gamma(\pi)} \mathbb{E}_{\mathbf{z} \sim \mathcal{D}(\mathbf{x})} \prod_{i=1}^{-1} \frac{\partial p_i(\mathbf{z})}{\partial p_{i-1}(\mathbf{z})} \tag{14}$$

$$= \mathbb{E}_{\mathbf{z} \sim \mathcal{D}(\mathbf{x})} \sum_{p \in \gamma(\pi)} \prod_{i=1}^{-1} \frac{\partial p_i(\mathbf{z})}{\partial p_{i-1}(\mathbf{z})} \tag{15}$$

$$= \mathbb{E}_{\mathbf{z} \sim \mathcal{D}(\mathbf{x})} \prod_{i=1}^{-1} \frac{\partial \pi_i(\mathbf{z})}{\partial \pi_{i-1}(\mathbf{z})} \tag{16}$$

**Algorithm 1:** Guided Pattern Refinement in the Embedding-level Graph (GPR-e)

---

**Result:** Significant Path $\pi^e$

initialization;

$\mathbf{x} \sim$ Input Tokens, $f \leftarrow$ BERT ;

$\mathcal{G}_e \leftarrow$ GetEmbeddingGraph($f$), $L \leftarrow$ GetNumberOfLayers($f$);

$N \leftarrow$ GetNumberOfTokens(f), $m \leftarrow$ GetIndex([MASK]), ;

$n_q \leftarrow$ GetQoINode($m, \mathcal{G}_e$), $n_w \leftarrow$ GetClfNode($\mathcal{G}_e$);

$\pi^e \leftarrow$ OrderedSet(), $\mathcal{C} \leftarrow \{\}$ ;

$j \leftarrow$ GetStartingIndex();                      `// The word we start the search with`

$n_j^0 \leftarrow$ GetNode($\mathcal{G}_w, \mathbf{h}_j^0$) ;                   `// Find the corresponding node`

$\pi^e \leftarrow$ Append($\pi^e, n_j^0$);

**for** $l \in \{1, ..., L - 2\}$ **do**

    $\mathcal{C} \leftarrow \{\}$;

    **for** $i \in \{0, ..., N - 1\}$ **do**

        $n_i^l \leftarrow$ GetNode($\mathcal{G}_w, \mathbf{h}_i^l$) ;

        $\pi_t \leftarrow$ Append($\pi, n_i^l$);

        $\mathcal{C} \leftarrow \mathcal{C} \cup \{\pi_t\}$ ;

    **end**

    $\pi^e \leftarrow \arg\max_{\pi' \in \mathcal{C}} \mathcal{I}(\mathbf{x}|\pi', \pi'_{-1} \rightarrow^{\mathcal{P}} n_q)$

**end**

$n_m^L \leftarrow$ GetNode($\mathcal{G}_w, \mathbf{h}_m^{L-1}$);

$\pi^e \leftarrow$ Append($\pi^e, n_m^L$);

$\pi^e \leftarrow$ Append($\pi^e, n_w$);

$\pi^e \leftarrow$ Append($\pi^e, n_q$);

---

# B   Appendix: Guided Pattern Refinement

## B.1   Algorithms for GPR

The pseudo code of the GPR algorithms are presented in algorithm 1 and 2.

## B.2   Optimality of GPR

The definition of pattern influence does not allow for polynomial-time searching algorithms such as dynamic programming. (Such an algorithm is possible for simple gradients/saliency maps but not for integrated gradients due to the expectation sum over the multiplication of Jacobians along all edges). As for the optimality of the polynomial-time greedy algorithm, we hereby include a statistical analysis by randomly sampling 1000 alternative patterns for 100 word patterns in the *SVA-obj* task and sentiment analysis task (SST2). The pattern influence of those random paths shows that the patterns extracted GPR are (1) more influential than 999.96 and 1000(all) random patterns, averaged across all 100 word patterns evaluated for embedding-level attention-level patterns, respectively for *SVA-obj*; the same holds for sentiment analysis (1000 and 1000); (2) The extracted pattern influences are statistically significant assuming all randomly sampled patterns' influences follow a normal distribution, with $p =$ 2e-10 for and $p = 0$ for embedding-level and attention-level patterns, respectively for *SVA-obj*; the same holds for sentiment analysis ($p = 0$ for and $p = 0$). In other words, the pattern influence values are far more significant than a random pattern, also confirmed by the high concentration values shown in Sec. 4.4.

# C   Appendix: Baseline Patterns

## C.1   Baseline: Attention-based baseline

We introduce the implementation details of attention-based patterns, inspired by [2] and [49]. Consider a BERT model with $L$ layers, between adjacent layers $l$ and $l + 1$, the attention matrix is denoted by $M_l \in \mathbb{R}^{A \times N \times N}$ where $A$ is the number of attention heads and $K$ is the number of embeddings.

**Algorithm 2:** Guided Pattern Refinement in the Attention-level Graph (GPR-a)

---

**Result:** Significant Path $\pi^a$

initialization;

$\mathbf{x} \sim$ Input Tokens, $f \leftarrow$ BERT ;

$\mathcal{G}_e \leftarrow$ GetEmbeddingGraph($f$), $\mathcal{G}_a \leftarrow$ GetAttentionGraph($f$);

$m \leftarrow$ GetIndex([MASK]);

$n_q \leftarrow$ GetQoINode($m, \mathcal{G}_e$);

$L \leftarrow$ GetNumberOfLayers($f$), $N \leftarrow$ GetNumberOfTokens(f) ;

$\pi^e \leftarrow$ GPR-e($\mathbf{x}, \mathcal{G}_e$);            `// Find embedding-level path first`

$\mathcal{C} \leftarrow \{\}$;

**for** $i \in \{0, ..., |\pi^e| - 2\}$ **do**

  **if** *ExistAttentionBlock($\pi_i^e, \pi_{i+1}^e$);*      `// Check if this is a Transformer Layer`

  **then**

      $\mathcal{A}_i \leftarrow$ GetHeadsBetween($\mathcal{G}_a, \pi_i^e, \pi_{i+1}^e$);  `// Get all attention heads and the`
      `skip connection node`

      $\pi_{head}^e \leftarrow$ Slice($\pi_0^e, \pi_i^e$);            `// Take a slice between two nodes`

      $(a_i^*, c_i^*) \leftarrow \arg\max_{(a_i, c_i) \in \mathcal{A}_i} \mathcal{I}(\mathbf{x}|\pi_{head}^e \cup \{a_i, c_i\}, c_i \to^{\mathcal{P}} n_q)$;

      $\mathcal{C} \leftarrow \mathcal{C} \cup \{(a_i^*, c_i^*)\}$;

  **else**

      continue;

  **end**

**end**

$\pi^a \leftarrow$ InsertNode($\pi^e, \mathcal{C}$);   `// Insert attention nodes into the embedding-level`
`path at the corresponding place`

---

Each element of $M_l[a, i, j]$ is the attentions scores between the $i$-th embedding at layer $l$ and the $j$-th embedding at layer $l + 1$ of the $a$-th head such that $\sum_i M_l[a, i, j] = 1$. We average the attentions scores over all heads to lower the dimension of $M$: $\tilde{M}_l \overset{\text{def}}{=} \frac{1}{A} \sum_a M_l[a]$. We define the baseline attention path as a path where the product of each edge in this path is the maximum possible score among all paths from a given source to the target.

**Definition 4 (Attention-based Pattern)** *Given a set of attention matrices $\tilde{M}_0, \tilde{M}_1, ..., \tilde{M}_{L-1}$, a source embedding $x$ and the quantity of interest node $q$, an attention-based pattern $\Pi_{attn}$ is defined as*

$$\Pi_{attn} \overset{\text{def}}{=} \{x, h_*^1, h_*^2, ..., h_*^{L-2}, q\}$$

*where*

$$h_*^1, h_*^2, ..., h_*^{L-2}$$
$$= \arg \max_{j_1, j_2, ..., j_{L-2}} P(h_{j_1}^1, h_{j_2}^2, ..., h_{j_{L-2}}^{L-2})$$
$$P = \tilde{M}_0[s, j_1] \tilde{M}_{L-1}[j_{L-2}, t] \prod_{l=1}^{L-2} \tilde{M}_0[j_l, j_{l+1}]$$

[2] considers an alternative choice $\hat{M}_l \overset{\text{def}}{=} 0.5I + 0.5\tilde{M}_l$ to model the skip connection in the attention block where $I$ is an identity matrix to represent the identity transformation in the skip connection, which we use GPR in Sec. 4, we use $\hat{M}_l$ to replace $\tilde{M}_l$ when returning the attention-based pattern. Dynamic programming can be applied to find the maximum of the product of attentions scores and back-trace the optimal nodes at each layer to return $h_*^1, h_*^2, ..., h_*^{L-2}$.

### C.2 Baseline: Conductance and Distributional Influence

We find patterns consisting of nodes that maximizes *condutance* [9] and *internal influence* [23] to build baseline methods $\Pi_{cond}$ and $\Pi_{inf}$, respectively. Definitions of these two measurements are shown as follows:

**Definition 5 (Conductance [9])** *Given a model $f : \mathbb{R}^d \to \mathbb{R}^n$, an input $\mathbf{x}$, a baseline input $\mathbf{x}_b$ and a QoI q, the conductance on the output $\mathbf{h}$ of a hidden neuron is defined as*

$$c_q(\mathbf{x}, \mathbf{x}_b, \mathbf{h}) = (\mathbf{x} - \mathbf{x}_b) \circ \sum_i \mathbf{1}_i \circ \mathbb{E}_{\mathbf{z} \sim \mathcal{U}}\left[\frac{\partial q(f(\mathbf{z}))}{\partial \mathbf{h}(\mathbf{z})} \frac{\partial \mathbf{h}(\mathbf{z})}{\partial z_i}\right] \tag{17}$$

*where $\mathbf{1}_i$ is a vector of the same shape with $\mathbf{x}$ but all elements are 0 except the $i$-th element is filled with 1. $\mathcal{U} := Uniform(\overline{\mathbf{x}_b\mathbf{x}})$.*

**Definition 6 (Internal Influence [23])** *Given a model $f : \mathbb{R}^d \to \mathbb{R}^n$, an input $\mathbf{x}$, a QoI q and a distribution of interest $\mathcal{D}$, the internal influence on the output $\mathbf{h}$ of a hidden neuron is defined as*

$$\chi_q(\mathbf{x}, \mathbf{h}) = \mathbb{E}_{\mathbf{z} \sim \mathcal{D}}\left[\frac{\partial q(f(\mathbf{z}))}{\partial \mathbf{h}(\mathbf{z})}\right] \tag{18}$$

In this paper, we use a uniform distribution over a path $c = \{\mathbf{x} + \alpha(\mathbf{x} - \mathbf{x}_b), \alpha \in [0,1]\}$ from a user-defined baseline input $\mathbf{x}_b$ (`[MASK]`) to the target input $\mathbf{x}$, reducing the internal influence to Integrated Gradient(IG) [39] if we multiply the $(\mathbf{x} - \mathbf{x}_b)$ with the distributional influence, as mentioned in Sec. 4. IG is an extention of Aumann Shapley values in the deep neural networks, which satisfies a lot of natural axioms: efficiency, dummy, path-symmetry, etc.[38]. Choices of DoIs, besides the one used in IG, include Gaussian distributions with mean $\mathbf{x}$ [37] and Uniform distribution around $\mathbf{x}$ [48] are shown to have other nice properties such as robustness against adversarial perturbations.

The relatively higher ablated accuracy of $\Pi_{\text{cond}}$ is expected: pattern influence (Def. 3) using our settings of $\mathcal{D}$ and QoI, with exactly one internal node, reduces to conductance[9], therefore picking most influential node per layer using conductance is likely to achieve similar patterns. However, this gap in ablated accuracy between $\Pi_{\text{cond}}$ with $\Pi_+^e$, albeit small, shows the utility of the GPR algorithm over a comparable layer-based approach.

### C.3 Baseline: Variation Statistics for random patterns

| Patterns | SVA | | | | RA | | SA |
|---|---|---|---|---|---|---|---|
| | Obj. | Subj. | WSC | APP | NA | GA | |
| $\Pi_+^e$ (ours) | **.96** | **.99** | **1.0** | **.96** | **.98** | **.99** | **.97** |
| $\Pi_{\text{rand}}^e$ | .55±.017 | .60±.014 | .55±.016 | .55±.016 | .55±.015 | .56±.014 | .52±.030 |
| $\Pi_{\text{attn}}^e$ | .61 | .55 | .49 | .63 | .56 | .65 | .47 |
| $\Pi_{\text{cond}}^e$ | .93 | .91 | .88 | .90 | .71 | .95 | .94 |
| $\Pi_{\text{inf}}^e$ | .66 | .71 | .50 | .56 | .68 | .50 | .49 |
| $\Pi_+^a$ (ours) | **.70** | **.62** | **.56** | **.65** | **.78** | **.89** | **.86** |
| $\Pi_{\text{rand}}^a$ | .50±.010 | .50±.008 | .50±.009 | .50±.011 | .50±.011 | .50±.008 | .49±.022 |
| $\Pi_{\text{repl\_skip}}^a$ | .50 | .58 | .54 | .52 | .55 | .50 | .48 |
| original | .96 | 1.0 | 1.0 | 1.0 | .83 | .73 | .92 |

Table 2: Ablated accuracies(Table 1 Left) of $\Pi_+$ and baseline patterns expanded with standard deviations for the random baseline patterns over 50 runs.

# D  Appendix: Experiment Details

## D.1  Task and Data details

For linguistics tasks, example sentences and accuracy can be found in Table 3. We also include the accuracy of a larger BERT model ($\text{BERT}_{\text{BASE}}$) which are comparable in performance with the smaller BERT model used in this work. First 5 tasks are sampled from [28](MIT license), the last task is sampled from dataset in [24](Apache-2.0 License), all datasets are constructed as an MLM task according to [13](Apache-2.0 License). In order to compute quantitative results, we sample sentences with a fixed length from each subtask. SA data (SST-2) are licensed under GNU General Public License. The BERT pretrained models[7] are licensed under Apache-2.0 License.

## D.2  Experiment Setup

Our experiments ran on one Titan V GPU with tensorflow[1]. On average GPR for attention-level and embedding-level take around half a minute to run for each instance in SVA and RA, and around 1

| Task | Type | Example | BERT Small | BERT Base |
|---|---|---|---|---|
| SVA | | | | |
| Object Relative Clause | SS SP PS PP | the author that the guard likes [MASK(is/are)] young | 1 0.92 0.9 1 | 1 0.96 0.98 1 |
| Subject Relative Clause | SS SP PS PP | the author that likes the guard [MASK(is/are)] young | 1 1 1 1 | 1 0.96 0.98 1 |
| Within Sentence Complement | SS SP PS PP | the mechanic said the author [MASK(is/are)] young | 1 1 1 1 | 1 1 1 1 |
| Across Prepositional Phrase | SS SP PS PP | the author next to the guard [MASK(is/are)] young | 1 1 0.98 1 | 0.99 0.98 0.98 1 |
| Reflexive Anaphora | | | | |
| Number Agreement | SS SP PS PP | the author that the guard likes hurt [MASK(himself/themselves)] | 0.66 0.66 0.83 1 | 0.6 0.74 0.83 0.96 |
| Gender Agreement | MM MF FF FM | some wizard who can dress our man can clean [MASK(himself/herself)] | 0.78 0.32 1 0.8 | 1 0.96 0.9 0.66 |

Table 3: Example of each agreement task and their performance on two BERT models.

min for SA, with 50 batched samples to approximate influence. The whole quantitative experiment across all tasks takes around 1 days cumulatively.

### D.3 Code submission

We will release the code publicly once the paper is published.

## E Appendix: More Visualizations of Patterns

### E.1 Example Visualizations

Figure 6, 7, 8 shows similar attractor examples from Figure 3a and 3b, in three other evaluated subtask: SVA-Subj, SVA-APP, RA-NA. We observe similar discrepancies between SP and PS within each subtask, with `that`, and `across` and `that` functioning as attractors, respectively. Figure 9 through 13 show example patterns of actual sentences in the SST2 dataset.

### E.2 Aggregated Visualizations

In this section, we show the aggregated visualization(Fig. 14 and 15 ) across all examples of each case in two subtasks (SVA-Obj & NA-GA) by superimposing the patterns of individual instances (e.g. Figure 3a and 3b), while adjusting the line width to be proportional to the frequency of flow across all examples. The words within parenthesis represent one instance of the word in that position. The aggregated graphs verify (1) generality of patterns across examples in each case (including SS and PP). (2) a more intuitive visualization of the pattern entropy in these two tasks, with RA-NA showing "messier" aggregated patterns, or larger entropy.

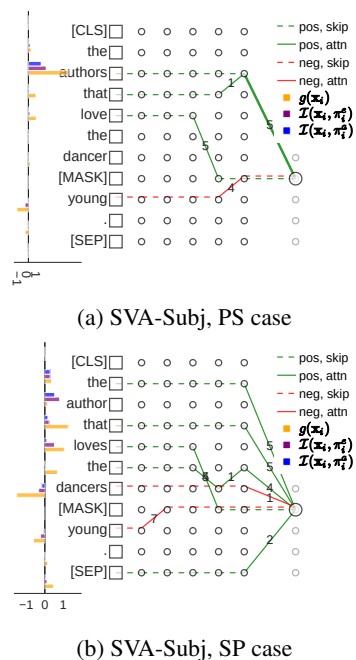

(a) SVA-Subj, PS case

(b) SVA-Subj, SP case

Figure 6: Examples of SVA-Subj

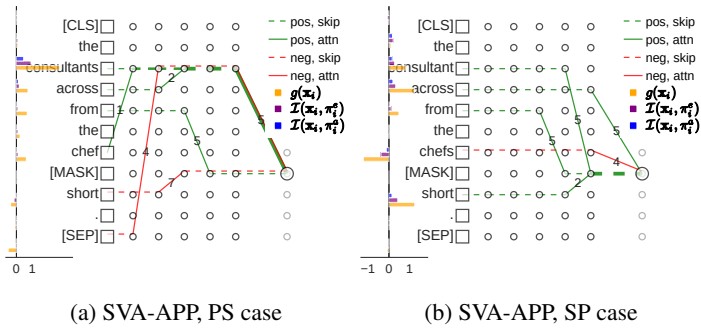

(a) SVA-APP, PS case        (b) SVA-APP, SP case

Figure 7: Examples of SVA-APP

### E.3 Impact Statement

Our work is expected to have general positive broader impacts on the uses of machine learning in the natural language processing. Specifically, we are addressing the continual lack of transparency in deep learning and the potential of intentional abuse of NLP systems employing deep learning. We hope that work such as ours will be used to build more trustworthy systems. Transparency/interpretability tools as we are building in this paper offer the ability for human users to peer inside the language models, e.g. BERT, to investigate the potential model quality issues, e.g. data bias and the abuse of privacy, which will in turn provide insights to improve the model quality. Conversely, the instrument we provide in this paper, when applied to specific realizations of language generation and understanding, can be used to scrutinize the ethics of these models' behavior. We believe the publication of the work is more directly useful in ways positive to the broader society. As our method does not use much resources for training or building new models for applications, we believe there are no significant negative social impacts.

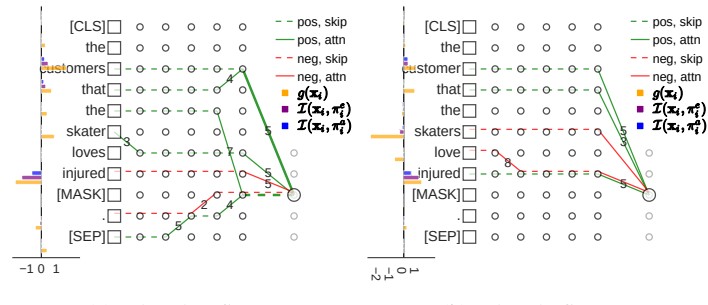

(a) RA-NA, PS case       (b) RA-NA, SP case

Figure 8: Examples of RA-NA

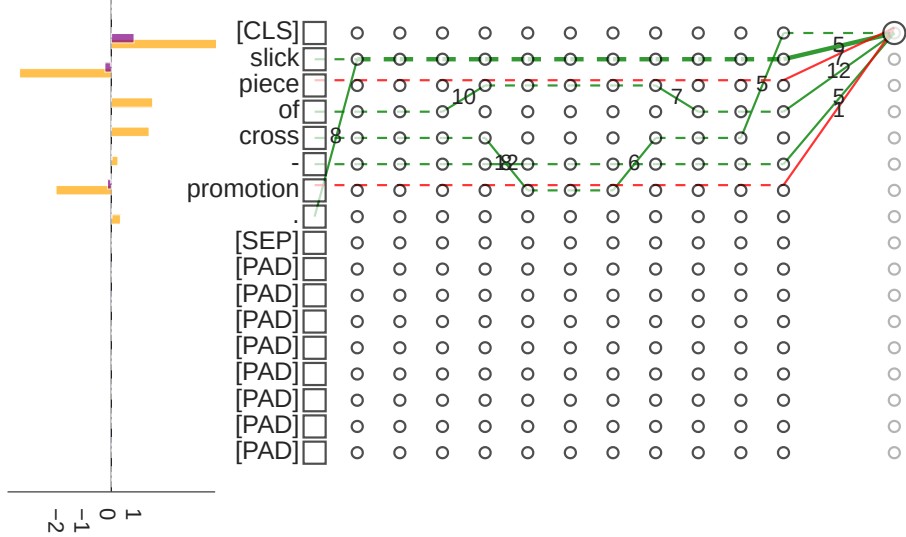

Figure 9: Example pattern of a positive sentence in SA

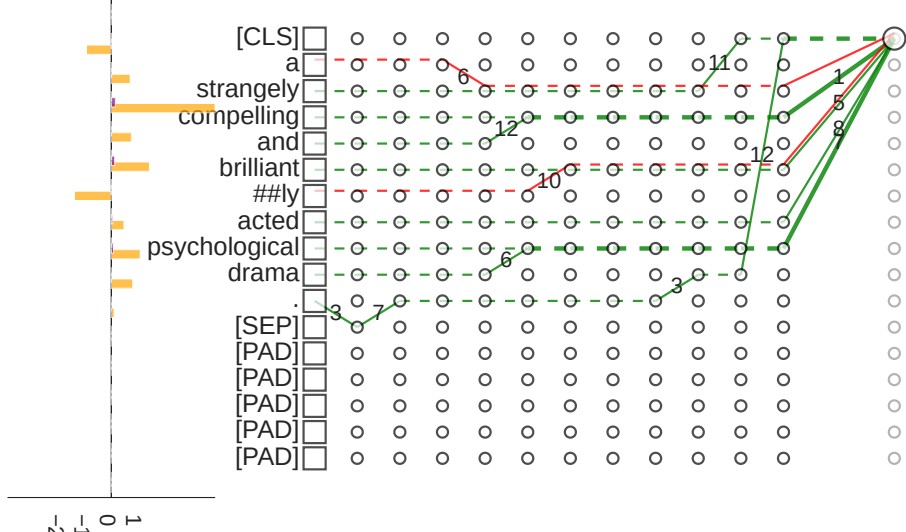

Figure 10: Example pattern of a positive sentence in SA

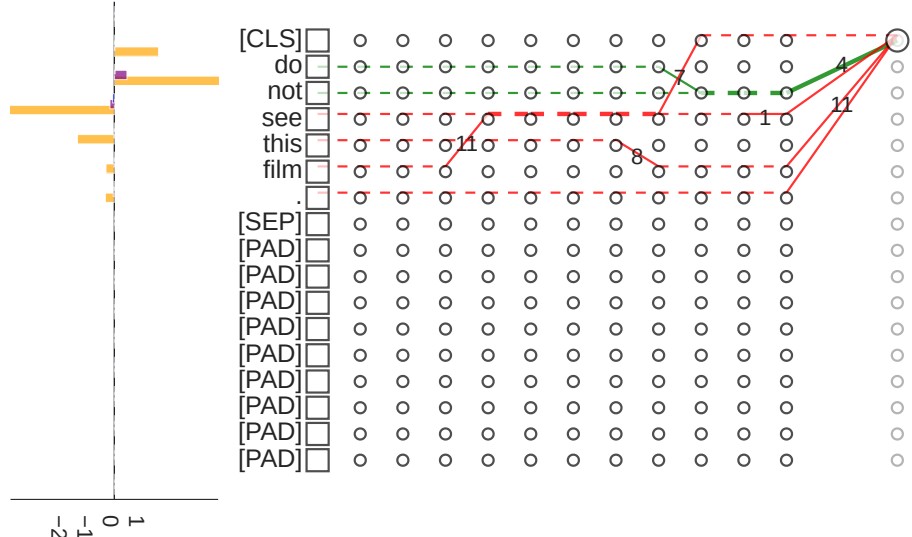

Figure 11: Example pattern of a negative sentence in SA

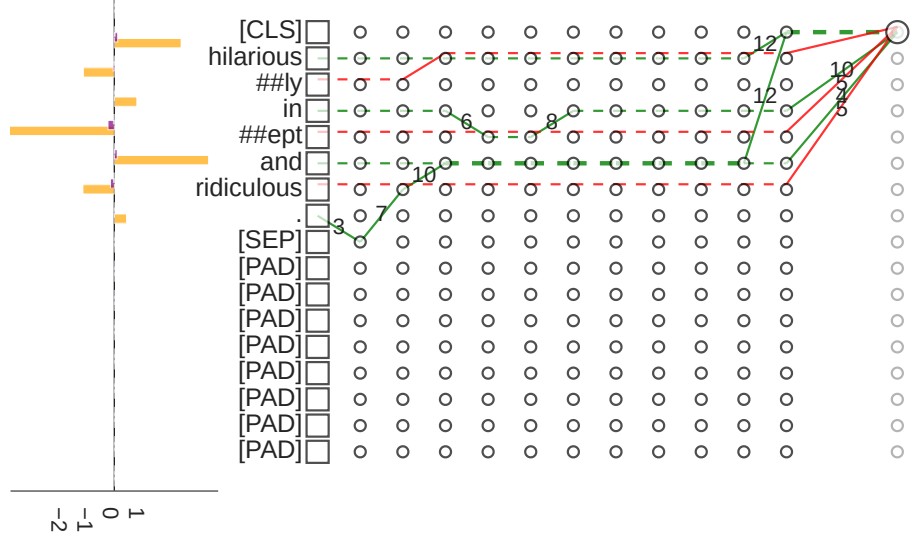

Figure 12: Example pattern of a positive sentence in SA

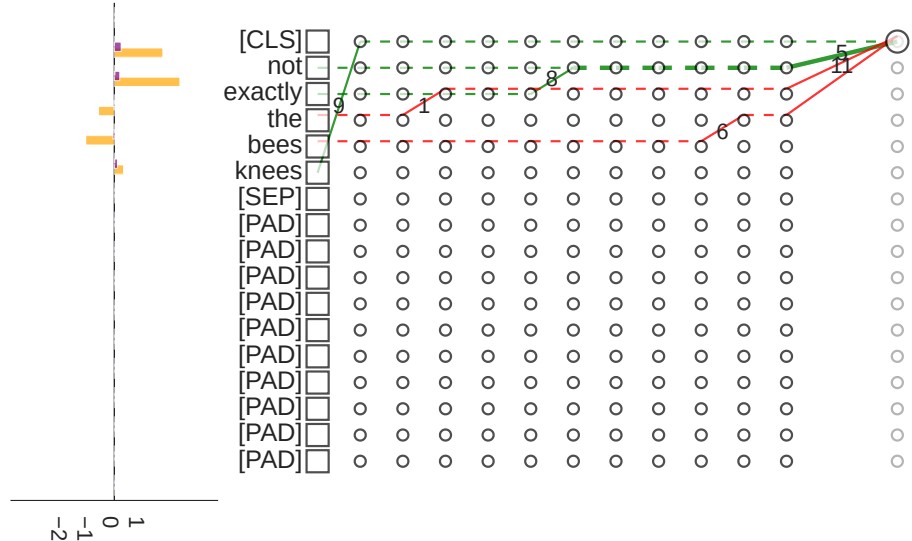

Figure 13: Example pattern of a negative sentence in SA

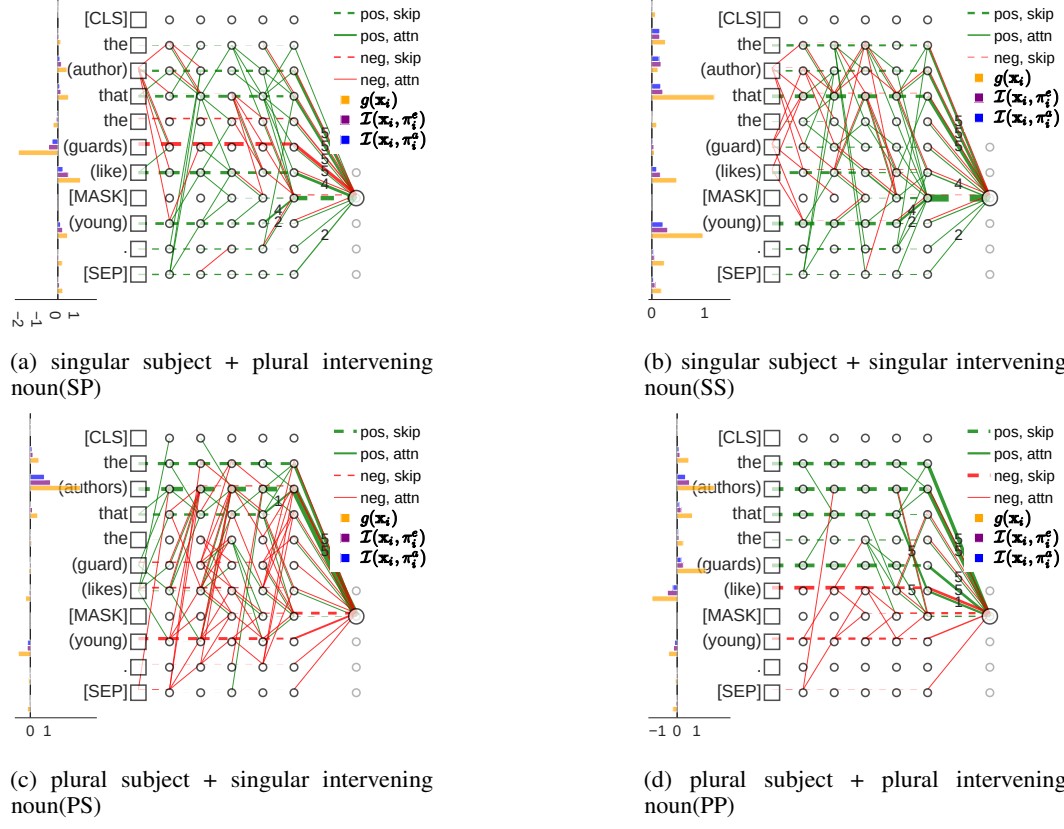

(a) singular subject + plural intervening noun(SP)

(b) singular subject + singular intervening noun(SS)

(c) plural subject + singular intervening noun(PS)

(d) plural subject + plural intervening noun(PP)

Figure 14: SVA-Obj. Aggregated

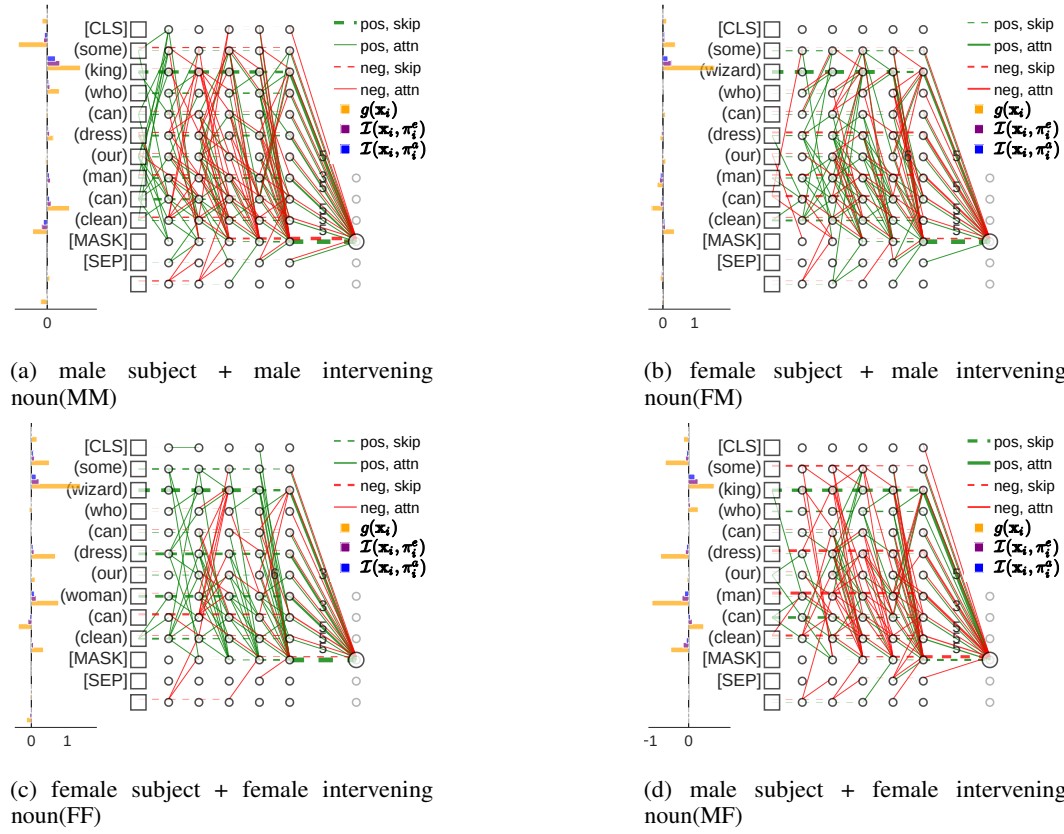

(a) male subject + male intervening noun(MM)

(b) female subject + male intervening noun(FM)

(c) female subject + female intervening noun(FF)

(d) male subject + female intervening noun(MF)

Figure 15: RA: GA, Aggregated