# OpenReview forum: "Influence Patterns for Explaining Information Flow in BERT"
_NeurIPS.cc/2021/Conference — NeurIPS 2021 Poster_

### Official Review · Reviewer_MnyX · 2021-07-05

**Rating:** 7
**Confidence:** 4

**Summary:**

As BERT achieves promising results in many NLU tasks (e.g., GLUE benchmark and SquAD benchmark), providing explanation for BERT has also received increasing attention recently. Some literature examines pre-trained BERT encoders, through using probes to analyze their semantic and syntactic knowledge. Some other work focuses on fine-tuned BERT, and uses gradient-based explanation (or variants of gradient) to provide local explanation for each input sentence. This work novelly proposes influence patterns, to abstract sets of paths through the BERT model, as to to illustrate the information flowing process within BERT. Experimental results shed light on the decision making process of BERT models.


**Limitations And Societal Impact:**

Yes, the limitations are adequately addressed by the authors.

**Main Review:**

Advantages:

1.The proposed method is novel. The abstracted paths could illustrate the information flowing process within BERT in an friendly manner to users.

2. The paper is well written and I enjoy reading it.

3. Lots of interesting findings are found and the findings are convincing to me. For example, skip connections are important, and this could explain why lots of attention heads could be pruned without influencing the model performance. The decision making logic for phrases “not good” and “not bad” is novel.

I have the following minor concerns.

1. The proposed GPR model seems to work well with input texts with short length. Will the model work with longer input and with a larger model (e.g., BERT-large)?

2. Do the findings work for more complex NLU tasks? For example, the finding that skip connection is more important compared to attention heads, could be attributed to the simplicity of the tasks examined in this work, e.g, SVA and RA. For more complex NLU tasks, such as MNLI and SquAD, does the finding still hold?

3. Although lots of works use sparsity to evaluate the explanation quality, I do not think this is a good metric.  Suppose that the underlying model to be explained using dense patterns to make decisions, then the provided sparse explanations would be low fidelity to its decision making process.


**Time Spent Reviewing:**

5 hours

---

> ### Author Response · Authors · 2021-08-11
> **Response to reviewer MnyX**
>
> 1. The algorithm in this paper, which is linear with respect to the number of layers or the number of words, can be easily applied to larger models.
>
> 2. We speculate that the findings from the skip connections will most likely hold for other more complicated tasks such as NLU, but possibly to a lesser extent as the author suggested. However, we do believe that skip connections are important model components which directly relays information that does not need contextualization from layer to layer, indicating that attention scores alone cannot explain the mechanism of the network.
>
> 3. The explanations should ideally be sparse for humans to understand. For the tasks evaluated in this paper, we found keeping one node per layer retains a large percentage of influence and ablated accuracy nevertheless(Table 1), therefore proving that the abstracted explanation is faithful to the model's encoding of the task-relevant concepts. We do agree with the author’s assessment that other representations in other NLP tasks could be more distributed therefore could flow through more paths in the network.  However, the GPR algorithm can be easily adapted to include more paths with heuristics such as pick top n nodes per layer.

---

### Official Review · Reviewer_pApo · 2021-07-06

**Rating:** 6
**Confidence:** 4

**Summary:**

In order to explain how information flows from input tokens to output predictions in transformer-style models such as BERT, this paper defined “influence patterns”, abstractions of sets of paths (of neural network nodes) through a transformer model. BERT_SMALL and BERT_BASE were used to perform the analysis under two groups of NLP tasks (1) subject-word agreement (SVA) and reflexive anaphora (RA); and (2) sentiment analysis (SA).

The observations include (1) a significant portion of information flows in BERT go through the residual skip connections, instead of multi-head (self-)attention heads; (2) BERT could use grammatically incorrect (e.g., let “not” to be negative, without considering its context such as “not good” vs. “not bad”); (3) “influence patterns” account for information flows in BERT on average of 74% and is 25% more accurate than prior attention-based and layer-based explanation methods.

The definition of “influence patterns” using the idea of node paths and the major observations are the major contributions of this paper.


**Ethical Concerns:**

No concerns.

**Limitations And Societal Impact:**

No related negative societal impacts of this work.
Also, if the work goes deeper, it can be a good example of explaining “how/why BERT and related large-scale pretraining language models work”.


**Main Review:**

Understanding why “attention is all you need” in transformer-style models such as BERT is one of the important directions in NLP and deep learning, especially for large-scale pretrained models.

However, one important reference is missing in this paper, “Attention is Not All You Need: Pure Attention Loses Rank Doubly Exponentially with Depth” (https://arxiv.org/abs/2103.03404 ) (opened Mar 5, 2021) where a quite similar finding of: skip connection also plays an important role compared with multi-head self-attentions. Also, in that paper, “paths” of nodes (skip connection and heads) were all included. That paper further gave a theoretical prove that why “skip connection” is essential and what will be the final consequence if transformer models without “skip connection”. Those results have a rather big overlap with the most important observation of this paper. Thus, would like to suggest the authors read that paper and make a clear “novelty/difference description”.


Another concern is that currently, BERT_SMALL and BERT_BASE are the two major models used for analyzing, taking three tasks and focused on MLM loss only. Since these two models only have 6 layers and 12 layers, it is also important to see the results in larger BERT models with L=24 and so on, and also with other GLUE/SuperGLUE tasks where sentence-order-prediction loss is required to capture paragraph-level linguistic information. “important information flow through skip connections 3 times more often than attentions” – is this conclusion/observation only on L=6 and L=12 BERT? Prefer deeper analysis of how the number of layers matter and if there is a trade-off or relation between L and importance of skip connections.

Detailed questions and comments:
1.	Line 10, as mentioned before, better mention existing path-based methods as well;
2.	Line 53, p_{-1}’s -1 is a bit ambiguous, possibly better mention that it’s the last element index of a list? (alike python?)
3.	Lines 61-63, we see “nodes”, “head nodes”, and “skip nodes”, possibly it is better to distinguish the nodes in one transformer encoder layer, such as multi-head self-attention, layer-norm, skip connection, and feed-forward layers’ nodes; also how they are selected for “path” construction. Some paths include only part of the nodes, if there are deeper reasons of this “skip”/”jump” paths, I would like to learn.
4.	Figure 1, “a, s, QoI” are used in the figure and the explaining of them are far away in the paper, possibly better give quick annotation of them in the caption;
5.	Figure 2, not quite understand why only one “node” is selected for analysis, since the connections are fully connected, I am wondering if ideas alike “beam” (top-N) best paths make sense to the final explanation;
6.	Lines 160-162, as mentioned before, possibly you can try L=24 or larger number of layers, or other types of pretraining tasks besides MLM (line 154);
7.	Line 219, I am wondering how many “not” included words/subwords were masked during MLM training? Does the masking techniques matter to this finding?
8.	Table 1, direct comparison of these quantitative results with baselines (attention/layer) is a bit difficult to follow. Are there any intuitive examples that show the merits of “influence patterns”? that is, I am wondering what will be the conclusions drawn by baselines (attention/layer) from your existing examples.
9.	Table 3 of appendix, is/are, himself/themselves are selected as examples here. Just wondering how often these words appeared in the original texts for training BERT? And any further comparison of their respective frequencies in the training/fine-tuning datasets?

Typos:
1.	Line 49, “are are” -> “are”;
2.	Line 150, “two group of” -> two groups of?
3.	Line 104, 105, since you used “1) firstly”, possibly you can only use 1) or only firstly;


**Time Spent Reviewing:**

20

---

> ### Author Response · Authors · 2021-08-11
> **Response to reviewer pApo**
>
> Response to concerns
> 1. We appreciate the reviewer for pointing out this concurrent work Dong et al. (per [NeurIPS policy](https://neurips.cc/Conferences/2021/PaperInformation/NeurIPS-FAQ) we believe this is a concurrent rather than a prior work). We were not aware of the preprint version of this paper and it was only recently published as a ICML 2021 conference paper after this submission. We would like to add discussions about this paper in the revision and we hereby share our preliminary comments regarding the novelty of this paper compared with Dong et al: Dong et al. provides a specific decomposition of self-attention modules to arrive at the finding about skip connection whereas our work is not limited to attention structures and should generalize to any modules as long as they are differentiable, i.e. self-attentions. We are happy to corroborate if our empirical findings agree with Dong et al.’s decomposition of self-attention.
>
> 2. Besides MLM we have included a sentiment analysis task which is fine-tuned to the SST-2 task (a GLUE task) which produces Finding IV in the paper and corresponding quantitative results in Table 1. For our analysis, there isn’t a significant difference between the importance of skip connections between 6 and 12 layers, we will include a detailed comparison in the final version of the paper.  We hypothesize that we should be able to find similar results about skip connections in a 24-layer BERT. Consider the attention head as the model component where the BERT contextualizes different embeddings; several figures (Fig 3a, 3b, 4a and 4b) show once an influence pattern goes through an attention head,  it is often followed by successive skip connections. This potentially illustrates that once the contextualization is done, few contextualizations are needed. However, we do speculate that the relative importance of skip connections compared with attention heads may vary depending on tasks and models. The algorithm in this paper, which is linear with respect to the number of layers or the number of words, can be easily applied to larger models.
>
> Response to detailed questions and comments:
>
> *1. Line 10, as mentioned before, better mention existing path-based methods as well;*
>
> Please see above.
>
> *2. Line 53, p_{-1}’s -1 is a bit ambiguous, possibly better mention that it’s the last element index of a list? (alike python?)*
>
> We will revise the notation in the final version.
>
> *3. Lines 61-63, we see “nodes”, “head nodes”, and “skip nodes”, possibly it is better to distinguish the nodes in one transformer encoder layer, such as multi-head self-attention, layer-norm, skip connection, and feed-forward layers’ nodes; also how they are selected for “path” construction. Some paths include only part of the nodes, if there are deeper reasons of this “skip”/”jump” paths, I would like to learn.*
>
> We will colorcode the nodes from different layers in the final version. The embedding nodes, head nodes are commonly studied architecturally in previous work therefore allow us to make a head-to-head comparison. However, one can choose to include any nodes to the computation graph,  such as layer-norm nodes, feed-forward nodes.  The skip “path” essentially captures the gradient when a node is directly copied between layers, which can be separated (chain rule in calculus) from the gradient flowing through the attention blocks from the same node. This decomposition provides a way to isolate the influence of skip connections vs attention blocks.
>
> *4. Figure 1, “a, s, QoI” are used in the figure and the explaining of them are far away in the paper, possibly better give quick annotation of them in the caption;*
>
> We will revise it in the final version.
>
> *5.Figure 2, not quite understand why only one “node” is selected for analysis, since the connections are fully connected, I am wondering if ideas alike “beam” (top-N) best paths make sense to the final explanation;*
>
> The expectation in the definition of influence does not allow for polynomial-time algorithms like A-star, BFS or beam search.(for example, a1*b1 + a2*b2) can not be broken down into f(a1,a2) and f(b1,b2), such an algorithm is possible for simple gradients/saliency maps, or attention-based graphs, but not for integrated gradients/smoothgrad). However, the GPR algorithm can be easily adapted to include more paths with heuristics such as pick top n nodes per layer. For the tasks evaluated in this paper, however, we found keeping one node per layer retains a large percentage of influence and ablated accuracy nevertheless(Table 1). We do agree with the author’s assessment that other representations in other NLP tasks could be more distributed therefore could flow through more paths in the network.
>
> *6.Lines 160-162, as mentioned before, possibly you can try L=24 or larger number of layers, or other types of pretraining tasks besides MLM (line 154);*
>
> Please see above .
>
> *7.Line 219, I am wondering how many “not” included words/subwords were masked during MLM training? Does the masking techniques matter to this finding? *
>
> We are using the original Bert, which is pretrained and released by Google and masks around 15% percent of words. We are not sure if the masking technique will change the influence patterns, but we speculate the primary reason for this discrepancy is the stronger correlation of “not” with negative sentiment in the training data during the fine tuning process.
>
> *8.Table 1, direct comparison of these quantitative results with baselines (attention/layer) is a bit difficult to follow. Are there any intuitive examples that show the merits of “influence patterns”? that is, I am wondering what will be the conclusions drawn by baselines (attention/layer) from your existing examples.*'
>
> We have found that other attention-based methods do not generate interpretable patterns as they are mostly random. Note that these methods are often used for attributing important nodes in each layer, instead of attributing a path of important nodes across all layers. Therefore, our method is the only viable method to show how information flows from input to output.
>
> *9. Table 3 of appendix, is/are, himself/themselves are selected as examples here. Just wondering how often these words appeared in the original texts for training BERT? And any further comparison of their respective frequencies in the training/fine-tuning datasets?*
>
> The BERT Model[Devlin 2019] are pretrained on bookscorpus and wikipedia dataset and not fine-tuned since we use it in a MLM setting. We are not sure of the frequency of these tokens but we speculate that they should be pretty common. There could potentially be a relative difference in frequencies,  and they might impact how the model prefers one over the other. However, this task/dataset was created to evaluate the grammatical understanding of English[Marvin 2019, Lin 2019], therefore from a grammatical standpoint, the model should choose the right answer regardless of its relative frequency. However, how the relative frequency of the words impacts the model’s decision could be an interesting topic for future work and can potentially be uncovered by the use of influence patterns.
>
> We will revise the typos in the final version.

---

> > ### Comment · Reviewer_pApo · 2021-09-02
> > **thanks for the detailed answers and responses**
> >
> > Thanks the authors for the detailed answers.
> >
> > (1) I see about the "attention is not all you need" work and I noticed that this paper itself appeared quite earlier than the paper published in ICML2021, thus I agree with the authors, that is a recurrent work. Also, just as the authors mentioned, hope to see some detailed difference comparison in the future version of this paper;
> >
> > (2) besides NLP sentiment anaylsis tasks, it will be wonderful to see some vertical domain analysis such as financial news' sentiment analysis and causal inferences to better understand the reald-world application connections of the conclusions from your paper.

---

### Official Review · Reviewer_NGL2 · 2021-07-16

**Rating:** 5
**Confidence:** 4

**Summary:**

This submission proposes a novel graph-based method to visualize the information model in transformer-based models such as BERT. The proposed method utilizes graph search algorithm to produce patterns which are generalized paths in graphs. The advantage of these patterns is that they are sparse and have fewer nodes than a conventionally defined path in the computation graph and are therefore potentially more interpretable. The authors demonstrate that such visualization can lead to interesting findings on several natural language processing tasks. Through quantitative evaluations, the authors also show that the proposed method is better than previous methods in terms of ablated accuracy and sparsity.

That being said, my main concern is that I doubt such visualization can bring much *practical values* for model development and model understanding. I detail my concerns below.

**Ethical Concerns:**

I do not have any ethical concerns with this work.

**Ethics Review Area:**

["I don’t know"]

**Limitations And Societal Impact:**

I do not see any negative societal impact from this work.

**Main Review:**



Pros:

- The proposed method is novel
- The submission has done a good job comparing the proposed method to previous visualization methods through both qualitative and quantitative studies.
- The paper is well-written and clear

Cons:

I find it difficult to translate the kind of insights revealed by such visualization to any knowledge that is beneficial for model development or our understanding of BERT. I'll illustrate by the examples discussed in section 4.2

1. Finding I: Skip connection matters. While I find it interesting to see this behavior shown up in the visualization, it is hardly surprising. I believe we can safely say that without skip connections, it would be very difficult train these transformers if not impossible. Such fact already tells us the importance of skip connection.
2. Finding II: First, I am not sure how much this finding is tied to these two specific instances. If the conclusion we want to have is that BERT may rely on ungrammatical correlations for prediciton, I think we need to have more observations than these three examples. Second, even if we can establish this conclusion, what does this tell us? Are there meaningful takeaways we can have to change our model.

Ultimately my position is that these visualizations should be means instead of ends in themselves. I understand that this comment may also be applied to many published previous work to this submission but I think it is important. I am happy to be convinced otherwise if the author can show other findings that can help us debug or improve our models.

**Time Spent Reviewing:**

6

---

> ### Author Response · Authors · 2021-08-11
> **Response to Reviewer NGL2**
>
> 1. We agree with the reviewer that skip connections are important for *training* neural network models. Our paper highlights that specific skip connections also matter for *explaining* neural network model predictions. This is in contrast to a body of work that focuses exclusively on attention heads, and ignores or does not isolate the role of skip connections in explaining model predictions [Clark 2019, Wu2020, Abnar2020]. Additionally, as opposed to the general statement of “skip connections matter”,  influence pattern can help identify which skip connections matter more than the nearby attention heads, on a local level, which is not attainable during the training process.
>
> 2. The purpose of this finding was not to identify at a large scale BERT's learning of ungrammatical concepts, but to demonstrate how influence patterns, as an explanation method, can help uncover the model’s ungrammatical encodings. However, in appendix E.2, we include an aggregated visualization by overlapping the visualizations from each evaluated instance to show that the trend observed in Figure 3 and 4 holds on a larger scale .
>
> As to the general comment of the paper, the focus of this paper is to show how the explanation capability can surface how BERT contextualizes various concepts, while debugging or improving the model on a larger scale is not within the scope of the paper. However,  this work can potentially serve as a building block for future exploration of model improvement and debugging.  Note that in Table 1 the first row shows that the ablated model retains almost all the original accuracy for SVA tasks, and increases the model’s performance for RA and SA tasks. Although the debugging implications of this table wasn’t the focus of the paper, it can be taken further to improve the model’s performance.
>
> We agree that explanations can be a means to other ends, such as model debugging. However, We also believe explainability can be an end in itself,  as demonstrated by previously published Neurips papers (ex: Dabkowski 2017, Wang 2020, Reif 2019, as cited in this paper). We believe explainability by itself builds trust and enables humans to work effectively with machine learning models in decisioning tasks.

---

### Official Review · Reviewer_LK45 · 2021-08-02

**Rating:** 7
**Confidence:** 3

**Summary:**

This paper introduces the concept of influence patterns which provides a novel way to quantify and analyse the flow of information through a sequence of model nodes in a neural network. Using influence patterns, the paper shows the end to end flow of information in a BERT architecture and experiments also show that the

**Limitations And Societal Impact:**

It would have been useful if authors included a discussion on the computational cost of finding the influence patterns.

**Main Review:**

The paper is well written and easy to follow. The proposed technique of quantifying the flow of information in a bert model using influence patterns seems to be novel enough and can be very useful in better understanding the bert models and potentially identifying any redundancies/shortcomings. Moreover, as the experiments show is significantly better than the baselines. However, I am a little puzzled by the results in last three columns of Table-1, correct me if I am wrong but how is the original model performance worse than the method where you just retain the nodes from \pi_+ and make everything else zero?
Did the authors also try using these influence patterns to prune the network by only retaining the important nodes and make the model inference and possibly even training faster?


**Time Spent Reviewing:**

2

---

> ### Author Response · Authors · 2021-08-11
> **Response to Reviewer LK45**
>
> The result may look surprising at first glance but can still be reasonable if explained as follows: the positive influence pattern is able to identify a subgraph responsible for making the prediction; therefore, when ablating less influential nodes(including the negatively influential ones which drive the prediction in the wrong direction), the subgraph performs better than the entire graph because negative and noisy signals are prevented from flowing from the input embeddings to the output.
>
> We haven’t tried to train the pruned network directly but we have done inference on the pruned models to get the ablation results in Table 1 Left. The main reason why we did not do so is that the ablation experiments shown in section 4 mainly serve as an evaluation instead of a guide to prune or to improve the model’s architecture. Pattern-based pruning & retraining of a network requires a lot more setups and comparison with a different set of baselines, which are not in the scope of this paper. However, we believe this work should serve as a building block for future exploration of model improvement.
>
> The computation cost is included in the appendix D. For each example, the algorithm takes around 30 seconds to 1 min on a Titan V GPU, depending on the length of the sentences. However, the time can be greatly reduced if we only limit the analysis to high-influence tokens.

---

> > ### Comment · Reviewer_LK45 · 2021-09-10
> > **Thanks for response**
> >
> > I am still not fully convinced because its not the case that the original model is always performing worse than the method where you just retain the nodes from \pi_+ and make everything else zero. However, i would trust the authors and believe that there is no error and the results are correct.
> > Given this i am increasing my rating to 7.

---

### Decision · Program_Chairs · 2021-09-27

**Decision:**

Accept (Poster)

**Comment:**

All reviewers agree that this paper proposes a novel method to understand functioning of Transformer models through influence patterns. 3/4 reviewers find the contributions interesting and substantial and voted for acceptance. One reviewer raised concerns about the purpose of such techniques and how they can help in design better architectures. Authors responded by saying such understanding is important for building trust and debugging. I completely agree with Authors on this and think such analysis studies are very important and am glad to recommend acceptance. I encourage Authors to include suggested changes/new comparisons brought up by reviewers in the final version.